# The Impact of Diet on the Fecal Microbiota Transplantation Success in Patients with Gastrointestinal Diseases—A Literature Review

**DOI:** 10.3390/nu17203314

**Published:** 2025-10-21

**Authors:** Natalia Komorniak, Katarzyna Gaweł, Anna Deskur, Jan Pawlus, Ewa Stachowska

**Affiliations:** 1Department of Human Nutrition and Metabolomics, Pomeranian Medical University in Szczecin, 71-460 Szczecin, Poland; natalia.komorniak@pum.edu.pl; 2Department of Gastroenterology, Pomeranian Medical University in Szczecin, 71-460 Szczecin, Poland; katarzyna.gawel@pum.edu.pl (K.G.); anna.deskur@pum.edu.pl (A.D.); 3Department of General Mini-Invasive and Gastroenterological Surgery, Pomeranian Medical University in Szczecin, 71-460 Szczecin, Poland; jan.pawlus@pum.edu.pl

**Keywords:** microbiota, dietary patterns, fecal microbiota transplantation, fiber, gastrointestinal diseases

## Abstract

**Background:** Fecal microbiota transplantation (FMT) is a therapeutic method involving the administration of appropriately prepared feces from a healthy donor to the gastrointestinal tract of a recipient. This literature review aims to summarize and critically evaluate the available evidence on the impact of different dietary patterns and nutrients on the efficacy of FMT. **Methods:** The present literature review focuses on the impact of diet on the gut microbiota in the context of the effectiveness of fecal microbiota transplantation. A literature review was conducted based on the PubMed Database. **Results:** More and more data confirm the close link between diet and gut microbiota and suggest that proper nutrition before and after FMT may support the effectiveness of this procedure. It appears that increased fiber intake significantly delays the loss of diversity in the transplanted microbiota, thereby enhancing the beneficial clinical effects following FMT. Additionally, the use of an anti-inflammatory components in the diet combination with FMT could be effective in achieving clinical remission in patients with ulcerative colitis. **Conclusions:** Based on the literature review, it appears that the most optimal nutritional model (through its beneficial effect on the composition of the gut microbiota, short-chain fatty acids production, and intestinal barrier integrity) to support the effectiveness of FMT is an anti-inflammatory diet rich in dietary fiber (for both the donor and the recipient).

## 1. Introduction

Fecal microbiota transplantation (FMT) is a therapeutic method involving the administration of appropriately prepared feces from a healthy donor to the gastrointestinal tract of a sick recipient [1]. The transplant aims to restore balance in the recipient’s gut microbiota by increasing microbial diversity, changing the composition of bacterial metabolites acting on the host organism, reducing intestinal barrier permeability, and improving the functioning of the gut–brain axis [2]. The donor’s gut microbiota is designed to compete with and displace harmful gut bacteria, thereby restoring the balance of the recipient’s gut microbiome. Additionally, there is a noticeable effect on the production of health-promoting short-chain fatty acids (SCFAs), whose presence is essential for maintaining the integrity of the intestinal barrier and modulating the immune system, thereby reducing inflammation and improving intestinal health [3]. This treatment method has been used in diseases caused by dysbiosis, especially in preventing recurrence of *Clostridioides difficile* infection (with a cure rate of up to 90%) [1,3]. Compared to traditional therapies, such as antibiotics, FMT and standardized microbiome therapies were highly effective in treating *Clostridium difficile* infection (rCDI), demonstrating high short-term efficacy and favorable long-term safety [4,5]. In inflammatory bowel diseases such as Crohn’s disease and ulcerative colitis, therapies targeting the gut microbiota are also useful. Due to the complex pathogenesis in IBD, this therapy requires precision in the selection of recipient and donor, diet, and xenobiotics, as well as consideration of interactions between strains [6]. Additionally, a recent umbrella review found potential benefits of FMT in the treatment of inflammatory bowel disease (IBD), particularly ulcerative colitis (UC). FMT demonstrated benefits in terms of clinical remission and endoscopic remission/response. Low methodological quality and variable certainty of evidence require the conduct of high-quality RCTs to strengthen the evidence [7]. Al-Habsi et al. report health benefits of FMT, including improved immune function, improved well-being, alleviation of symptoms associated with irritable bowel disease, reduced allergy severity, and antibacterial and anti-inflammatory effects [8]. Promising data also relate to research focusing on the potential use of FMT in the treatment of diseases such as obesity, diabetes, metabolic syndrome, autoimmune diseases, inflammatory bowel diseases, and skin diseases [3]. It is worth noting that the effectiveness of FMT is influenced by many factors, including donor characteristics, recipient factors, and procedural aspects of the FMT protocol [9].

While several studies suggest that a high-fiber or Mediterranean diet promotes microbial diversity and increases the effectiveness of FMT, other studies indicate minimal or no effect of dietary interventions on clinical outcomes. Furthermore, many studies differ in terms of design, methods of dietary assessment, and duration of follow-up, making direct comparison difficult. To date, there are no standard dietary recommendations for patients undergoing FMT, either before or after the procedure, and nutritional factors are often overlooked in clinical protocols [10,11]. Given these inconsistencies and the growing awareness that diet is a modifiable factor influencing immunity and gut microbiota function [12], a comprehensive summary of current research findings is needed. This review therefore aims to summarize and critically evaluate the available evidence on the impact of different dietary patterns and nutrients on the efficacy of FMT. By identifying gaps in knowledge and areas of convergence, we hope to provide a rationale for incorporating tailored nutritional strategies into future FMT protocols.

## 2. Methods

The present literature review focuses on the impact of diet on the gut microbiota in the context of the effectiveness of fecal microbiota transplantation. In particular, we focused on studies of different dietary models in FMT donors and recipients in the context of gastrointestinal disorders. A literature review was conducted based on the PubMed Database. The keywords were checked and combined for the following terms: fecal microbiota transplantation, diet, fiber, gastrointestinal diseases, constipation, diarrhea, colitis ulcerosa, Crohn’s disease, irritable bowel syndrome, inflammatory bowel disease, Mediterranean diet, Western diet, anti-inflammatory diet. Studies that were not in the English language, letters to editors and abstracts to conferences were excluded.

### 2.1. The Impact of Microbiota on Health and the Impact of Diet on Microbiota

The term “microbiota” includes a huge number of microorganisms, including bacteria, archaea, viruses, fungi, and protozoa that colonize the human digestive tract [13,14]. Many studies conducted in recent years indicate that the gut microbiota plays a key role in maintaining health, but its abnormal composition may contribute to the development of many diseases [15]. This is because these microorganisms perform many key functions related to maintaining homeostasis, including participating in digestion, preventing colonization by pathogens, modulating the immune system [16,17], produce short-chain fatty acids, which strengthen the intestinal barrier and reduce inflammation in the body [18].

It is worth noting that the gut microbiota is unique to each individual, and its condition is influenced by both endogenous factors (e.g., age) and exogenous factors (e.g., diet, physical activity, medication, stress) [19,20]. It seems that diet is one of the main factors influencing the composition and functioning of the gut microbiota. Dietary components affect intestinal transit time, mucin and digestive compound secretion (e.g., bile, enzymes), inflammation, oxidative stress, and the functioning of the immune and nervous systems and the intestinal barrier [21]. The Western diet (rich in highly processed foods, high in animal protein, sugar, salt, saturated fatty acids, and low in dietary fiber) contributes to the degradation of the mucus layer in the intestine, promoting bacterial translocation (and thus susceptibility to infection and inflammation), as well as a reduction in the number of beneficial bacteria of the genus *Bifidobacterium* and *Eubacterium* [22,23]. In turn, the Mediterranean diet model (rich in fiber, vegetables, fruits, legumes, nuts, olive oil, and moderate consumption of meat, fish, and dairy products) [24] influences the induction of anti-inflammatory processes that increase host immunity [25,26], increases the content of SCFA in feces [27], and also increases the number of health-promoting bacteria of the genera *Faecalibacterium*, *Roseburia* and *Ruminococcus* [28].

### 2.2. Requirements for Stool Donors

Scientific studies report that FMT recipients experience changes in their microbiological profiles and shifts in the composition of their gut microbiota towards a profile similar to that of the donor, and that these changes are observed for at least six months [29]. For this reason, it seems extremely important to properly prepare and select a stool donor. The stool donor should be a healthy person, free from infectious diseases and other chronic illnesses, who is able to donate material frequently if necessary [30]. The transplant material can be obtained from people in the recipient’s circle or from a stool bank. In both cases, the potential donor must meet certain criteria and undergo appropriate tests [31].

The donor must be thoroughly examined, which includes collecting a medical history (the questions in the questionnaire concern infectious diseases, risky behaviors, and health status, with a focus on intestinal health) and performing laboratory tests, including blood tests (e.g., screening for viral hepatitis, liver enzyme tests, C-reactive protein, or a full blood count with differentia), stool tests (including screening for *Clostridioides difficile* toxin B, *Campylobacter*, *Salmonella* and *Shigella*, Shiga toxin-producing *Escherichia coli*, norovirus and rotavirus, *Helicobacter pylori* antigen) [30,31,32]. After passing the screening, the patient may be considered a stool donor. Each patient should have a specific period for donating stool to the bank. However, this period cannot exceed 3 months, and after the end of the donation period, the donor undergoes further testing after 4 weeks to rule out the possibility of a serological window. Each active donor should undergo periodic evaluation every 3 months. Stool should be delivered to the processing center within 2 h of defecation. If the time between defecation and processing exceeds 30 min, it should be stored under refrigerated conditions to avoid changes in its microbiological composition. After the stool is delivered to the stool bank, the process of processing the stool and preparing it for transplantation begins [31,33].

### 2.3. Indications, Risks, and Procedure for Conducting FMT

Current guidelines recommend FMT for the prevention of recurrent *Clostridioides difficile* infection (CDI) after two recurrences, with cure rates approaching 90%. The use of FMT in the management of severe and fulminant CDI, resulting in decreased mortality and colectomy rates. In refractory CDI patients who are not candidates for surgery, FMT may be a salvage/rescue therapy [1,34].

Abnormalities in the microbiome may play an important role in the pathogenesis of inflammatory bowel disease. Randomized controlled trials (RCTs) in ulcerative colitis have established fecal microbiota transplantation (FMT) as an effective therapy [35,36,37]. In one of the largest studies, symptom remission following FMT in patients with active Crohn’s disease was achieved in 57% of patients. A second FMT after 4 months was shown to maintain symptoms. Further randomized, placebo-controlled trials with long-term follow-up are necessary to maintain symptoms after the initial FMT [38,39]. Studies have shown a positive effect of FMT on reducing symptoms in patients with IBS, within three months 89.1% of patients achieved a response to treatment, the effectiveness increases with the dose [40].

Some authors suggest that gut microbiota transplantation may be a promising treatment for small intestinal bacterial overgrowth (SIBO) patients. Following FMT, bacterial diversity significantly increased in these patients, which may have contributed to the reduction in the severity of gastrointestinal symptoms in this group of patients [41]. More, carefully conducted research is needed in this area. In contrast, contraindications for FMT include:Relative contraindications: recent gastrointestinal surgery, severe acute illness, pregnancy and lactation, pediatric patients, elderly patients;Absolute contraindications: severe immunocompromise, gastrointestinal obstruction, toxic megacolon, recent major surgery, gastrointestinal tract perforation [3].

Once the decision is made to undergo intestinal microbiota transplantation, the patient should be appropriately prepared. The patient with *Clostridioides difficile* infection should be placed on a light diet, take the antibiotic vancomycin or fidaxomicin orally for 10 to 14 days, and implement sanitary procedures (washing, disinfection). Antibiotic therapy should be discontinued 24 h prior to FMT. A bowel cleansing agent, such as macrogol, should be taken 24 h prior to FMT, and a strict diet should be followed for 24 h and two hours after FMT administration. The patient can take fluids and necessary medications. There are two types of FMT preparations: a ready-made kit with a suspension of intestinal microbiota for administration through intraduodenal/intragastric/enteral tubes, gastroscope, colonoscope, percutaneous endoscopic gastrostomy (PEG), rectal enema, or capsules in a double coating (acid-resistant and enteric, released at the junction of the small intestine and large intestine for oral administration). A meta-analysis of studies on the effectiveness of FMT indicates that capsules are as effective as FMT performed via colonoscopy [42,43,44]. One jar of capsules is used within 15 min, the entire treatment in 1 day, optimally 2 h). For patients receiving FMT through the upper gastrointestinal tract, oral administration of a proton pump inhibitor (PPI) in standard doses twice daily is recommended 24 h prior to FMT. Treatment should be continued for up to two days after the procedure. An hour before FMT, an antiemetic medication, loperamide, should be administered in cases of very rapid transit [45].

A retrospective cohort study analyzed the long-term safety of FMT as a treatment for recurrent *Clostridioides difficile* infection compared to patients treated with a fixed bacterial mixture, i.e., rectal bacteriotherapy (RBT). Overall survival, risk of hospital admission, occurrence of certain pre-specified comorbidities (cancer, diabetes, hypertension, and inflammatory bowel disease), and risk of diagnosis of multidrug-resistant organisms were assessed. At five years post-treatment, there were no differences in survival, risk of hospital admission, or occurrence of any of the analyzed comorbidities. FMT was not associated with increased mortality, risk of hospital admission, or post-treatment morbidity compared with RBT [46]. Additionally, it should be noted that there are various routes of administration from recipient to donor, including oral administration in capsule form during colonoscopy. The dose, frequency of administration, and whether the donor material was fresh or frozen are also important considerations. Crothers et al. [47] reported that daily administration of encapsulated oral FMT (cFMT) capsules may prolong the persistence of FMT-induced changes in gut bacterial community structure and that there may be a link between cytokine production by MAIT cells and clinical response to FMT in UC populations [47]. Oral administration of frozen cFMT capsules is a promising delivery system for FMT and may be preferred for long-term treatment strategies in UC and other chronic diseases, but further evaluation will require consideration of home storage. Larger studies are needed to investigate the benefits of cFMT and determine its long-term impact on the colonic microbiome [47]. Murgiano et al. confirm the positive/beneficial effect of FMT in the treatment of inflammatory bowel disease. However, they point to certain limitations, including the lack of control groups, small sample size, and donor variability, which leads to variable study results. The type of FMT, route of administration (e.g., nasogastric tube, jejunal endoscopy, pouchoscopy, enemas), antibiotic preparation, infusion frequency, and donor selection are all important factors. A meta-analysis demonstrated clinical remission, which correlated with increased microbiome diversity and stability. Multiple infusions of FMT were observed to yield higher response rates than single infusions. FMT use should also be considered for the potential transfer of not only bacteria but also viruses, which can interact with the host and influence immune responses and metabolic functions. It seems that immunocompromised patients are particularly at higher risk [48].

Currently, FMT is generally considered a safe and effective form of treatment. The most common side effects of the therapy include mild symptoms (constipation, abdominal pain, nausea, vomiting, flatulence, febrile episodes). However, it is important to monitor the patient’s condition regularly, as a small number of people have also experienced serious side effects of the therapy, including sepsis, aspiration pneumonia, and intestinal perforation [49].

### 2.4. Scientific Research in the Context of Diet and the Effectiveness of FMT in Gastrointestinal Problems

Currently, recommendations regarding FMT do not include information on dietary recommendations for donors and transplant recipients. Meanwhile, an international survey of gastroenterologists clearly indicates that, in their opinion, diet (for both donors and recipients) is crucial to the effectiveness of FMT, and also highlights the lack of guidelines and the need for further research on diet around transplantation [10].

It is worth noting that, to date, research on the impact of diet on the effectiveness of FMT is very limited. In recent years, several studies have been published focusing on the potential impact of diet on the effectiveness of FMT in gastrointestinal disorders (Table 1). However, it should be remembered that, given that these studies were relatively short-term and involved a small number of participants, the results should be treated as a potential direction for further research. Some of the articles we found confirm the close link between diet and gut microbiota and suggest that proper nutrition before and after FMT may support the effectiveness of this procedure—however, these relationships are not yet fully understood, and the mechanisms require further research [11]. A study conducted on an animal model showed that FMT from different donors combined with increased dietary fiber can lead to different patterns of gut microbiota composition compared to combining FMT with a low-fiber diet. The researchers concluded that dietary fiber may play a more crucial role in shaping the composition of the microbiota than the FMT donor, and thus that fiber-based strategies may have a beneficial effect on the effectiveness of FMT in the recipient [50]. The important role of dietary fiber in the effectiveness of FMT has also been confirmed by studies involving humans. One study showed that increasing fiber intake (pectin supplement) in FMT recipients significantly delayed the loss of diversity of the transplanted microbiota and increased the beneficial clinical effects compared to FMT alone [51]. In patients with chronic constipation, combining FMT with soluble fiber supplementation resulted in a more significant improvement in bowel movement frequency and consistency compared to FMT alone (both in the short and long term) [52]. It also turns out that in UC patients, combining FMT with a parallel anti-inflammatory diet is more effective in inducing clinical remission than standard pharmacological therapy [53]. Another study showed that a 14-day dietary intervention in stool donors resulted in increased microbial diversity in recipients, changes in composition resembling the donor’s microbiota, and a significant reduction in calprotectin in the stool of recipients with ulcerative colitis. Similar changes were not observed in patients who received FMT from donors without dietary intervention [54]. For patients with irritable bowel syndrome, one of the most commonly used dietary models is the low FODMAP (fermentable oligosaccharide, disaccharide, monosaccharide, and polyol) diet [55]. Combining this diet with FMT in patients with diarrhea-predominant IBS produced better and more lasting results in alleviating symptoms than FMT alone. At 6 months, the clinical response rate was 62.5% for FMT + diet vs. 27.5% for FMT alone. It is also worth noting that the quality of life of patients was significantly higher in the FMT + diet group [56].

### 2.5. Dietary Recommendations for Donors and Recipients

Current clinical protocols for stool transplantation emphasize, above all, the importance of selecting suitable donors who have avoided antibiotic use and of properly cleansing the recipient’s intestines. However, no formal standard guidelines have been established regarding the diet of either donors or recipients of transplanted material [10,34,59]. Nevertheless, there is growing evidence that diet composition has a significant impact on the quality of the microbiota of both donors and recipients, as well as on the acceptance of the transplant by the recipient. A healthy microbiome is usually characterized by high taxonomic diversity, enriched with Bifidobacterium, bacteria of the species *Faecalibacterium prausnitzii*, *Akkermansia muciniphila*, and other butyrate-producing taxa, and a reduced frequency of pro-inflammatory *Proteobacteria*. In recipients, a diet rich in fiber increases the production of short-chain fatty acids, especially butyrate, which supports epithelial barrier integrity and immune regulation [11,50,60]. The combination of resistant starch and fructooligosaccharides has an additive effect on butyrate synthesis in the intestine [61]. Fermented foods such as yogurt, kefir, sauerkraut, or fermented soybeans provide live microorganisms and bioactive metabolites, increasing the diversity of microorganisms, providing *lactobacilli* and *bifidobacteria*, and jointly inhibiting the growth of pathogens in the intestines, which brings clinical benefits for people with irritable bowel syndrome or inflammatory bowel disease [62,63]. Foods rich in polyphenols (berries, pomegranates, green tea, cocoa, and olive oil) also promote the growth of *Akkermansia muciniphila* and *F. prauzsnitzii* bacteria, supporting anti-inflammatory and antioxidant effects, and fermentation further increases their bioavailability [62,64]. In addition, regular meals in accordance with the time of day, which means higher calorie intake in the early hours of the day, is associated with improved host metabolism and stabilization of the circadian rhythms of its microbiota [65].

In the case of donors, although the main guidelines (e.g., AGA or Open Biome) do not impose the use of specific nutritional regimens, observational and interventional data indicate that a diet rich in fiber, fermented products, and plants containing polyphenols is most conducive to maintaining a diverse, butyrate-synthesizing microbiota with a beneficial immune profile [34,61,64]. In contrast, a Western-style diet, rich in saturated fats, refined sugars, and ultra-processed foods, is associated with reduced microbial diversity, decreased SCFA synthesis, the development of pro-inflammatory taxa, and the gradual disappearance of the mucin layer that builds the intestinal barrier [66,67]. In turn, adherence to the Mediterranean diet is consistently associated with greater microbial diversity, enrichment of SCFA-producing *Bacterioidetes* and *Clostridium* bacteria, reduced *Proteobacteria* abundance, and improved intestinal barrier and immune function [68,69].

Dietary components that support a healthy and diverse gut microbiome include:Fiber fermentable to SCFA—resistant starch, inulin, arabinoxylans—provide substrates for saccharolytic fermentation, increasing butyrate levels and supporting the integrity of the mucosal membrane and immune barrier. Sources of these fibers include whole grains, legumes, bananas, onions, leeks, and chicory root [70].Fermented foods such as yogurt, kefir, sauerkraut, kimchi, and miso provide live microorganisms and bioactive metabolites. They increase the diversity of microorganisms and enrich *Lactobacillus* and *Bifidobacterium*, while reducing the number of pathogens [61].Products rich in polyphenols, such as berries, green tea, cocoa, and olive oil, stimulate the growth of beneficial taxa (*A. muciniphila*, *F. prusnitzii*), reduce oxidative stress, and work synergistically with fiber to increase butyrate production [64,70].The Mediterranean diet emphasizes vegetables and fruits, fish, legumes, whole grains, and extra virgin olive oil, which is consistently associated with greater microbial diversity and an anti-inflammatory metabolic profile of bacteria [68,69].Chrono-nutrition—regular meals with higher energy intake in the early part of the day improve metabolic homeostasis and microbial stability in the circadian cycle [65].

The worst dietary patterns for the microbiota are:A Western-style diet rich in saturated and trans fats, refined sugars, and highly processed foods. These foods reduce microbial diversity, inhibit efficient SCFA synthesis, promote the growth of pro-inflammatory taxa (*Enterobacetriaceae*), and damage the intestinal barrier, exacerbating endotoxemia [66,67].A diet low in fiber—limits substrates for saccharolytic fermentation, changing the metabolism of the microbiota towards proteolytic fermentation with the accumulation of potentially harmful metabolites such as ammonia or p-cresol [67,71].Excessive consumption of alcohol and artificial sweeteners is associated with dysbiosis, a reduction in the number of butyrate-producing bacteria, and impairment of the intestinal barrier [72,73].

## 3. Discussion

The available literature indicates that diet plays an important role in shaping the gut microbiota and can significantly influence the success of FMT. However, research results remain inconsistent. While several studies confirm [51,52,53] that a diet rich in fiber, such as the Mediterranean diet or anti-inflammatory diet, increases microbial diversity and promotes microbiota engraftment, other studies have not observed a clear improvement in clinical outcomes or long-term microbial stability [57]. These discrepancies are likely due to methodological differences between studies, including small sample sizes, differences in dietary assessment tools, and differences in FMT protocols and follow-up duration [10,11]. Despite these inconsistencies, converging evidence suggests that a nutrient-rich, fiber- and plant-based diet can create an intestinal environment conducive to the colonization of beneficial taxa such as *Faecalibacterium prausnitzii* and *Akkermansia muciniphila*, known for their anti-inflammatory and barrier supporting properties [50,64]. In contrast, a Western diet characterized by high consumption of processed fats, refined sugar, and highly processed foods is associated with lower microbial diversity and reduced short-chain fatty acid synthesis, which may negatively affect efficacy of FMT [66,67] (Figure 1). Future research should aim to standardize the assessment of dietary variables in FMT studies and to define specific dietary recommendations for both donors and recipients. Large-scale randomized controlled trials combining dietary intervention with stool transplantation are needed to clarify whether specific dietary patterns, in particular the Mediterranean diet, can improve transplant parameters. It also seems important to conduct research to determine the safety of FMT in the long term, taking into account potential confounding factors (e.g., medications, chronic diseases, infections). In summary, current scientific evidence emphasizes that diet remains an underappreciated but highly modifiable factor influencing FMT outcomes. The inclusion of personalized nutritional strategies, especially those emphasizing plant-based that diet remains an underappreciated but highly modifiable factor influencing FMT outcomes. The inclusion of personalized nutritional strategies, especially those emphasizing plant-based foods rich in fiber and polyphenols and an anti-inflammatory diet, can significantly improve microorganism engraftment and patient response. These insights provide a conceptual framework for the development of standard dietary recommendations that will accompany future FMT protocols.

## 4. Conclusions

Based on current literature, it appears that the most optimal nutritional model for both FMT donors and recipients is the Mediterranean diet, which is characterized by high consumption of dietary fiber, unsaturated fats, and plant products rich in polyphenols. This nutritional model supports the growth and metabolic activity of beneficial taxa in the microbiome, including *Faecalibacterium prausnitzii*, *Akkermansia muciniphila*, and *Bifidobacterium* spp., species that are associated with increased production of short-chain fatty acids and proper intestinal barrier integrity. The abundance of fiber and prebiotic substrates in this diet promotes interactions between commensal bacteria, maintaining the diversity and stability of microorganisms, which are key factors for successful microbiota transplantation after FMT. In addition, the Mediterranean diet has anti-inflammatory and immunomodulatory effects due to its content of omega-3 fatty acids and polyphenols, which can modulate host-microbiota interactions, reduce intestinal permeability, and limit the ex-pression of pro-inflammatory cytokines. These mechanisms may create a more favorable intestinal environment by increasing the persistence and functional integration of transplanted microorganisms. Despite these promising observations, evidence from controlled clinical trials remains limited. Future studies should aim to establish standard nutritional guidelines prior to FMT, establish the interaction between the donor, the recipient and the diet, and clarify how specific food components or bioactive compounds affect the engraftment of microorganisms and clinical outcomes.

## Figures and Tables

**Figure 1 nutrients-17-03314-f001:**
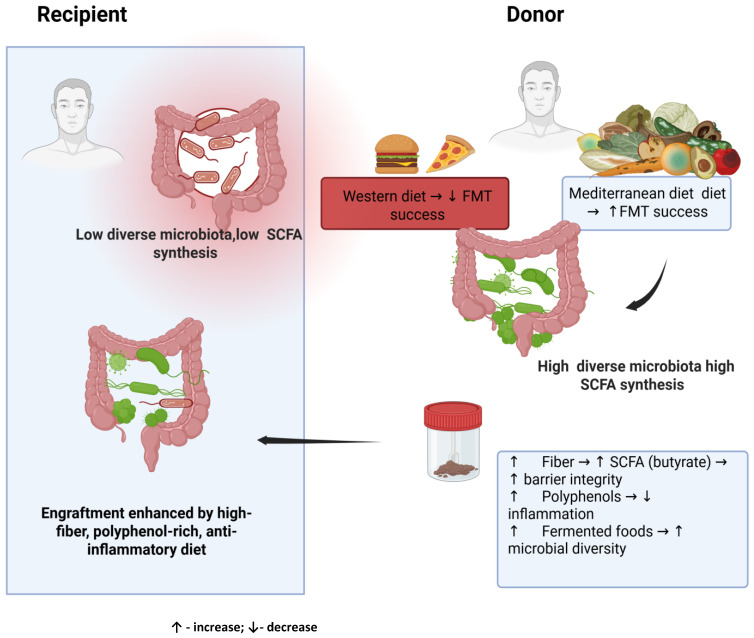
The role of the donor’s diet in preparing material for transplantation.

**Table 1 nutrients-17-03314-t001:** Summary of studies on the effectiveness of combining FMT with diet in people with gastrointestinal problems (adults, last 10 years).

Author	*n*	Study Design	Condition	Intervention	Outcomes	Study Limitations
[57]	27	Randomized 4-arm trial	UC (mild to moderate)	A single FMT or placebo with or without psyllium fiber supplementation for 8 weeks (recipient)	Single-dose FMT demonstrated clinical efficacy for UC compared to placebo but revealed no benefit of fiber supplementation.	Small study group Short study duration
[53]	66	Open-label RCT	UC (mild to moderate)	FMT + AID vs. SMT	FMT + AID had better induction (65.7 % vs. 35.5 %) and maintained deep remission at 48 weeks (25% vs. 0%) than SMT.	No FMT alone arm Small study group (FMT + AID − 35, SMT − 31)
[51]	20	RCT	UC (mild to moderate)	FMT vs. FMT + pectin	Pectin decreased the Mayo score by preserving the diversity of the gut flora following FMT for UC and enhanced the effect of FMT.	Small study group
[52]	29	Clinical Trial	Slow transit constipation	FMT + pectin	FMT in combination with soluble dietary fiber (pectin) had both short-term (4 week) and long-term (1 year) efficacy in treating constipation	No FMT alone arm Small study group
[58]	19	Randomized pilot study	CD with malnutrition	EEN + timing of WMT (WMT-Day1 = 8, WMT-DAY8 = 11)	EEN + immediate WMT improved the nutritional status and induced clinical remission in malnourished CD patients.	No FMT-alone arm Very short study duration (15 days) Small study group
[56]	80	A retrospective analysis of single-arm open-label prospective study	IBS with predominant diarrhea	FMT alone vs. FMT + LFD	LFD enhanced the efficacy of FMT, increased the gut microbial diversity after FMT, and strengthened the inhibitory effect of FMT on conditional pathogens.	Small study group (FMT alone = 40, FMT + LFD = 40)
[54]	21	Comparative Study	UC	Group 1: FMT alone Group 2: FMT with donors’ dietary pre-conditioning and UCED for the patients	FMT from diet conditioned donors followed by the UCED led to microbial alterations associated with favorable microbial profiles which correlated with decreased fecal calprotectin	Small study group
[40]	165	Randomized, double-blind, placebo-controlled study	IBS	Group 1: placebo Group 2: 30 g FMT Group 3: 60 g FMT Super donor with healthy diet and dietary supplements rich in proteins, vitamins, fiber and minerals)	FMT is more successful than placebo in curing IBS	Short study duration (1 month)
[10]	18	Observational Pilot Study	IBS, IBD undergoing FMT	Basic dietary education (by nurse) + high fiber diet (minimum 30 g/day) + inulin and pectin supplements + FMT	Higher diet quality associated with better outcomes post-FMT	Small study group High heterogeneity of the group (IBS = 7, UC = 4, CD = 7)

FMT—fecal microbiota transplantation, RCT—randomized controlled trial, AID—anti-inflammatory diet, SMT—standard medical therapy, EEN—exclusive enteral nutrition, WMT—washed microbiota transplantation, IBS—irritable bowel syndrome, IBD—inflammatory bowel disease, UC—Ulcerative colitis, CD—Crohn’s disease, LFD—low fodmap diet, UCED—UC exclusion diet.

## Data Availability

Not applicable.

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
