# Peer review of "The Impact of Diet on the Fecal Microbiota Transplantation Success in Patients with Gastrointestinal Diseases—A Literature Review"

_nutrients, 2025, doi:10.3390/nu17203314_

Round 1
Reviewer 1 Report
Comments and Suggestions for Authors
Dear authors,
I understand that you try to review the current literature on the way to reinforce the effectiveness of fecal microbial transplantation [FMT] or in other words to find the way to keep microbial diversity of the transplant as long as possible. Brilliant idea!
However,
your manuscript is a total of 5 pages plus a long table. Two and a half pages are introduction about the way of donor, the fecal preparation and the indications and risks of the procedure - all, to my opinion, being not necessary or not supported by the manuscript title.
Then, the main body [1.4] is only 29 lines [158-185] where you present in a few words the findings of 8 papers [ref 40 to 47], suggesting to read for detailed data the TABLE 1, where 10 papers presented!
Finally, chapter 1.5 deals with dietary recommendations for donors and recipients, which are, practically, recommendations for healthy life.
Thus, I suggest:
- to extensively shorten/delete the chapters 1.2 and 1.3
- to extensively analyze the chapter 1.4, dividing the articles according to the etiology of transplantation, as well as the urgent or not need for FMT. The same division for the Table.
- to comment on all of them
- to insert a section on Material/Method, referred to what data bases you search, the criteria of search, the articles found, the articles excluded and why and so on
Author Response
Reviewer 1
Dear authors,
I understand that you try to review the current literature on the way to reinforce the effectiveness of fecal microbial transplantation [FMT] or in other words to find the way to keep microbial diversity of the transplant as long as possible. Brilliant idea!
However,
your manuscript is a total of 5 pages plus a long table. Two and a half pages are introduction about the way of donor, the fecal preparation and the indications and risks of the procedure - all, to my opinion, being not necessary or not supported by the manuscript title.
Then, the main body [1.4] is only 29 lines [158-185] where you present in a few words the findings of 8 papers [ref 40 to 47], suggesting to read for detailed data the TABLE 1, where 10 papers presented!
Finally, chapter 1.5 deals with dietary recommendations for donors and recipients, which are, practically, recommendations for healthy life.
Thus, I suggest:
- to extensively shorten/delete the chapters 1.2 and 1.3
- to extensively analyze the chapter 1.4, dividing the articles according to the etiology of transplantation, as well as the urgent or not need for FMT. The same division for the Table.
- to comment on all of them
- to insert a section on Material/Method, referred to what data bases you search, the criteria of search, the articles found, the articles excluded and why and so on
Author’s response: Thank you very much for your comments. We have added a paragraph entitled “Methods” in which we describe how we analyzed the scientific literature. Additionally, in response to your valuable comments and those of other reviewers, we have decided to revise the paper. We focused on the impact of diet on the effectiveness of FMT in gastrointestinal disorders and diseases, emphasizing that the amount of research in this area is very limited, and given that diet is one of the main modulators of the microbiota, we believe that it is definitely worth conducting more research on this topic. In addition, we added a discussion and information about potential future perspectives to the article. We believe that the procedures for FMT donors and recipients are so important that we decided not to shorten the information contained therein. However, we expanded the data on long-term safety and potential factors affecting it. We have modified the contents of the table and rewritten the conclusions from the literature review. We hope that these changes will contribute to a better understanding of our article.

Reviewer 2 Report
Comments and Suggestions for Authors
Journal: NUTRIENTS
Manuscript ID
nutrients-3910848
Type: Review
Title: The impact of diet on the gut microbiota in the context of FMT and transplant success.
Authors: Natalia Komorniak , Katarzyna Gaweł , Anna Deskur , Jan Pawlus , Ewa Stachowska *
Section: Prebiotics and Probiotics
Special Issue: Probiotics, Postbiotics, Gut Microbiota and Gastrointestinal Health
___________________________________
OVERALL COMMENTS
Abstract
Based on the knowledge that fecal microbiota transplantation (FMT) is a therapeutic method involving a healthy donor to the gastrointestinal tract of a sick recipient. Since the effectiveness of FMT is influenced by many factors, including donor characteristics, recipient factors, procedural aspects of the FMT protocol, and that diet (as one of the key modulators of the microbiota) may be significant in the context of FMT efficacy, the authors of this study aimed to determine the potential impact of diet on the effectiveness of FMT.
TITLE
- The title should include the type of the review.
- The study include Ulcerative colitis and Crohn's disease. I suggest that the authors include these conditions in the title.
_______
ABSTRACT
- I suggest modifying the Abstract. Please include a clear objective;
- Please include the type of the review.
KEYWORDS
The authors presented the following keywords:
“microbiota; diet; fecal microbiota transplantation; mediterranean diet; western diet; Clostridium Difficile; Ulcerative colitis; Crohn’s disease; proton pump inhibitors; dysbiosis”
I suggest a change: “Keywords: microbiota; dysbiosis; diet; fecal microbiota transplantation; Clostridium Difficile; Ulcerative colitis; Crohn’s disease”
I suggest:
“digital health intervention; personalized healthy breakfast guidance; breakfast behavior and body composition”.
INTRODUCTION
- This section cites only 4 references. This number is insufficient. I suggest citing at least 10-15 articles that discuss FMT. There are many articles in databases such as PUBMED. There are articles from 2024 and 2025. Please include them.
- As examples:
10.1038/s41522-025-00808-5.
10.7759/cureus.90737
10.7759/cureus.90614
- The study appears to place some emphasis on ulcerative colitis and Crohn's disease. The authors could cite some information in the Introduction.
- This section covers many general concepts about microbiota and diet; however, I would appreciate reading a bit more about the inconsistencies in the literature and the rationale behind constructing a review of this type.
SUGGESTIONS FOR THE OTHER SECTIONS OF THE MANUSCRIPT
- Indications, risks, and procedures of FMT;
- I also miss a critical discussion of the long-term safety of FMT (risk of transferring pathogenic microorganisms or antimicrobial resistance genes). It also fails to sufficiently explore the methodological heterogeneity of the trials (differences in route, dose, frequency);
- I suggest including donor selection criteria;
- Regarding the results presented in Table 1:
-Many of the studies cited are pilot or open-label, with small patient numbers. In some cases, there was no control group (e.g., only FMT alone). This methodological weakness is not sufficiently highlighted.
-Furthermore, there is an inconsistency in the outcomes assessed (insulin sensitivity, clinical remission, quality of life), which makes comparisons difficult.
CONCLUSION
- This section is too short, The authors have much to conclude. Please review and improve this section.
____________
FUTURE PERSPECTIVES
I suggest including a section for Future Perspectives. The authors could comment that the findings of this study highlight the need for future research. As examples:
1- Conduct randomized, multicenter clinical trials with a larger sample size (most studies combining diet and FMT are still pilot or small). It is essential to conduct robust RCTs that confirm the synergistic effects of different dietary patterns on the success of FMT.
2- Standardize dietary protocols for donors and recipients.
3- Currently, there are no formal guidelines. Future research should compare dietary models.
4- Consider the role of individual variability.
5- Assess how factors such as genetics, age, prior antibiotic use, metabolic comorbidities, and baseline microbiota profile influence the interaction between diet and FMT.
6- Assess long-term safety.
7- Assess whether specific diets enhance the effects of FMT on metabolic (obesity, diabetes), cardiovascular, neurological, and autoimmune diseases.
_____________
REFERENCES
- Please see my comments in the Introduction section.
Author Response
Reviewer 2
OVERALL COMMENTS
Abstract
Based on the knowledge that fecal microbiota transplantation (FMT) is a therapeutic method involving a healthy donor to the gastrointestinal tract of a sick recipient. Since the effectiveness of FMT is influenced by many factors, including donor characteristics, recipient factors, procedural aspects of the FMT protocol, and that diet (as one of the key modulators of the microbiota) may be significant in the context of FMT efficacy, the authors of this study aimed to determine the potential impact of diet on the effectiveness of FMT.
TITLE
- The title should include the type of the review.
- The study include Ulcerative colitis and Crohn's disease. I suggest that the authors include these conditions in the title.
Authors’response: Thank you very much for your opinion. As suggested, the title has been modified to reflect the fact that the article is a literature review. The title now includes information that the analyzed data will mainly concern gastrointestinal diseases (however, we did not limit ourselves to Crohn's disease and ulcerative colitis; in the article, we also presented information on the possibility of using FMT in cases of constipation, IBS, or C. difficile infection).
ABSTRACT
- I suggest modifying the Abstract. Please include a clear objective;
- Please include the type of the review.
Authors’ Response: We would like to thank the reviewer for this opinion. We have rewritten the abstract in accordance with the following structure: introduction, methods, results, conclusions. In the abstract, we have included information that the article is a literature review, as well as the purpose of the literature analysis. We hope that these changes will contribute to a better understanding of our article.
KEYWORDS
The authors presented the following keywords: “microbiota; diet; fecal microbiota transplantation; mediterranean diet; western diet; Clostridium Difficile; Ulcerative colitis; Crohn’s disease; proton pump inhibitors; dysbiosis” I suggest a change: “Keywords: microbiota; dysbiosis; diet; fecal microbiota transplantation; Clostridium Difficile; Ulcerative colitis; Crohn’s disease” I suggest: “digital health intervention; personalized healthy breakfast guidance; breakfast behavior and body composition”.
Author’s response: Thank you very much for your insightful comment regarding the use of keywords. Based on Medical Subject Headings (MeSH), we have modified the keywords, which we hope will improve the visibility of our article in databases.
INTRODUCTION
- This section cites only 4 references. This number is insufficient. I suggest citing at least 10-15 articles that discuss FMT. There are many articles in databases such as PUBMED. There are articles from 2024 and 2025. Please include them. As examples: 10.1038/s41522-025-00808-5, 10.7759/cureus.90737, 10.7759/cureus.90614
- The study appears to place some emphasis on ulcerative colitis and Crohn's disease. The authors could cite some information in the Introduction.
- This section covers many general concepts about microbiota and diet; however, I would appreciate reading a bit more about the inconsistencies in the literature and the rationale behind constructing a review of this type.
Author’s response: We appreciate the reviewer's insight. The identified publications from 2024 and 2025 have been added to the paper, and the number of cited articles discussing FMT has been increased, as suggested by the reviewer. We also added information about inconsistencies in the literature, and the aim of making this review. We believe that these changes will improve/enrich our work.
SUGGESTIONS FOR THE OTHER SECTIONS OF THE MANUSCRIPT
- Indications, risks, and procedures of FMT;
- I also miss a critical discussion of the long-term safety of FMT (risk of transferring pathogenic microorganisms or antimicrobial resistance genes). It also fails to sufficiently explore the methodological heterogeneity of the trials (differences in route, dose, frequency);Ania
Author's Response: Thank you very much for this comment. At the reviewer's suggestion, we have emphasized the long-term safety of FMT (the risk of transferring pathogenic microorganisms or antimicrobial resistance genes) in the text. We have taken into account the methodological heterogeneity of the studies (differences in route of administration, dose, and frequency). The article also contains information on indications, transplantation procedures, and associated risks. We believe that these changes will contribute to a better understanding of our work.
- I suggest including donor selection criteria
Author's Response: The information about donor selection criteria is included in section “Requirements for stool donors”.
- Regarding the results presented in Table 1:
-Many of the studies cited are pilot or open-label, with small patient numbers. In some cases, there was no control group (e.g., only FMT alone). This methodological weakness is not sufficiently highlighted.
-Furthermore, there is an inconsistency in the outcomes assessed (insulin sensitivity, clinical remission, quality of life), which makes comparisons difficult.
Author’s Response: Thank you very much for this suggestion. We have revised the table, focusing only on studies concerning the combination of diet and FMT in gastrointestinal diseases, and emphasizing the limitations (mainly small study groups and short duration) of the studies.
CONCLUSION
This section is too short, The authors have much to conclude. Please review and improve this section.
Author’s Response: As suggested by the reviewer, we have rewritten the conclusions of our article.
____________
FUTURE PERSPECTIVES
I suggest including a section for Future Perspectives. The authors could comment that the findings of this study highlight the need for future research. As examples:
1- Conduct randomized, multicenter clinical trials with a larger sample size (most studies combining diet and FMT are still pilot or small). It is essential to conduct robust RCTs that confirm the synergistic effects of different dietary patterns on the success of FMT.
2- Standardize dietary protocols for donors and recipients.
3- Currently, there are no formal guidelines. Future research should compare dietary models.
4- Consider the role of individual variability.
5- Assess how factors such as genetics, age, prior antibiotic use, metabolic comorbidities, and baseline microbiota profile influence the interaction between diet and FMT.
6- Assess long-term safety.
7- Assess whether specific diets enhance the effects of FMT on metabolic (obesity, diabetes), cardiovascular, neurological, and autoimmune diseases.
Author’s response: Thank you very much for your valuable feedback. We agree that including future perspectives in the article may have a positive impact on the development of new research concepts. This information has been added to the “Discussion” section.
_____________
REFERENCES
Please see my comments in the Introduction section.
Author’s Response: In accordance with the reviewer's suggestion in the introduction section, we have expanded on some of the themes in the text, thereby increasing the number of articles cited in the text.

Reviewer 3 Report
Comments and Suggestions for Authors
Overall evaluation:
This review aim to determine the potential impact of diet on the effectiveness of Fecal microbiota transplantation. This review belongs to the category of clinical nutrition, but the content of the review is relatively simple and major revisions are recommended.
Specific modification suggestions:
1.Line2, the title should not appear unusual abbreviations, and it is suggested to change "FMT" to "Fecal microbiota transplantation".
2.Line13, the font "Fecal" does not need to be bold.
3.Line13-26, the writing of the Abstract part needs to be greatly revised, and should be written according to four parts: purpose, method, result and conclusion.
4.Line27-29, some Keywords have repeated expressions, suggesting that "microbiota and diet" be deleted.
5. The subtitle setting of the review is unreasonable. There are only two level 1 titles in the full text, namely "1. Introduction" and "2. Conclusions", which is unreasonable and the content of the review should be set as level 1 title.
6.Line31-54, the Introduction content is relatively simple, which should explain the necessity of carrying out this review and the research idea of this review.
7. There are too many words in Table 1. It is necessary to refine the contents of the column "Intervention and Outcomes" in the table.
8.Line248-255, the serial number "1,2,3" is used incorrectly, and the primary serial number cannot be used below the secondary serial number "1.6".
9. It is recommended that the review include a discussion section.
10. Pictures should be added to increase the appeal of the paper.
Author Response
Reviewer 3
Overall evaluation:
This review aim to determine the potential impact of diet on the effectiveness of Fecal microbiota transplantation. This review belongs to the category of clinical nutrition, but the content of the review is relatively simple and major revisions are recommended.
Specific modification suggestions:
1.Line2, the title should not appear unusual abbreviations, and it is suggested to change "FMT" to "Fecal microbiota transplantation".
Author's Response: It was corrected.
2.Line13, the font "Fecal" does not need to be bold.
Author's Response: It was corrected.
3.Line13-26, the writing of the Abstract part needs to be greatly revised, and should be written according to four parts: purpose, method, result and conclusion.
Author's Response: We would like to thank the reviewer for this opinion. We have rewritten the abstract in accordance with the following structure: introduction, methods, results, conclusions. In the abstract, we have included information that the article is a literature review, as well as the purpose of the literature analysis. We hope that these changes will contribute to a better understanding of our article.
4.Line27-29, some Keywords have repeated expressions, suggesting that "microbiota and diet" be deleted.
Author's Response: Thank you very much for your insightful comment regarding the use of keywords. Based on Medical Subject Headings (MeSH), we have modified the keywords, which we hope will improve the visibility of our article in databases.
- The subtitle setting of the review is unreasonable. There are only two level 1 titles in the full text, namely "1. Introduction" and "2. Conclusions", which is unreasonable and the content of the review should be set as level 1 title.
Author's Response: Thank you very much for this comment. We have removed the incorrectly applied numbering.
6.Line31-54, the Introduction content is relatively simple, which should explain the necessity of carrying out this review and the research idea of this review.
Author’s Response: Thank you very much for that opinion. As suggested, we have rewritten the “introduction” paragraph. We also added information about inconsistencies in the literature, and the aim of making this review. We believe that these changes will improve/enrich our work.
- There are too many words in Table 1. It is necessary to refine the contents of the column "Intervention and Outcomes" in the table.
Author’s Response: Thank you very much for this suggestion. We have revised the table, focusing only on studies concerning the combination of diet and FMT in gastrointestinal diseases, and emphasizing the limitations (mainly small study groups and short duration) of the studies.
8.Line248-255, the serial number "1,2,3" is used incorrectly, and the primary serial number cannot be used below the secondary serial number "1.6".
Author's Response: Thank you very much for this comment. We have removed the incorrectly applied numbering.
- It is recommended that the review include a discussion section.
Author’s Response: As suggested by the reviewer, we have added a paragraph entitled “Discussion” to the article. In this paragraph, we have also included potential future prospects for conducting scientific research in the context of the effectiveness and safety of FMT. We hope that the inclusion of this information will contribute to a better understanding of our literature review.
- Pictures should be added to increase the appeal of the paper.
Author’s Response: The scheme has been added to the article.

Round 2
Reviewer 1 Report
Comments and Suggestions for Authors
Dear authors,
I have to comment that instead of making changes according to my comment you adjusted your manuscript to a new title!
I have no more comments
Reviewer 2 Report
Comments and Suggestions for Authors
Dear authors,
Thank you very much for performing the modifications I suggested.
Reviewer 3 Report
Comments and Suggestions for Authors
Accept in present form.